# Signal and Texture Features from T2 Maps for the Prediction of Mild Cognitive Impairment to Alzheimer’s Disease Progression

**DOI:** 10.3390/healthcare9080941

**Published:** 2021-07-26

**Authors:** Alejandro I. Trejo-Castro, Ricardo A. Caballero-Luna, José A. Garnica-López, Fernando Vega-Lara, Antonio Martinez-Torteya

**Affiliations:** 1Escuela de Ingeniería y Ciencias, Tecnológico de Monterrey, Monterrey 64849, Mexico; a00818219@itesm.mx; 2Programa de Ingeniería Biomédica, Universidad de Monterrey, San Pedro Garza García 66238, Mexico; ricardo.caballerol@udem.edu (R.A.C.-L.); jose.garnical@udem.edu (J.A.G.-L.); fernando.vega@udem.edu (F.V.-L.); 3Departamento de Ingeniería, Universidad de Monterrey, San Pedro Garza García 66238, Mexico

**Keywords:** ADNI, Alzheimer’s disease, mild cognitive impairment, MRI biomarkers, signal, T2 maps, texture

## Abstract

Early detection of Alzheimer’s disease (AD) is crucial to preserve cognitive functions and provide the opportunity for patients to enter clinical trials. In recent years, some studies have reported that features related to the signal and texture of MRI images can be an effective biomarker of AD. To test these claims, a study was conducted using T2 maps, a sequence not previously studied, of 40 patients with mild cognitive impairment (MCI) from the Alzheimer’s Disease Neuroimaging Initiative database, who either progressed to AD (18) or remained stable (22). From these maps, the mean value and absolute difference of 37 signal and texture imaging features for 40 contralateral pairs of regions were measured. We used seven machine learning methods to analyze whether, by adding these imaging features to the neuropsychological studies currently used for diagnosis, we could more accurately identify patients who will progress to AD. The predictive models improved with the addition of signal and texture features. Additionally, features related to the signal and texture of the images were much more relevant than volumetric ones. Our results suggest that contralateral signal and texture features should be further investigated as potential biomarkers for the prediction of AD.

## 1. Introduction

Neurodegenerative diseases are a common and growing cause of mortality and morbidity, in which structural and chemical changes in the nervous system lead to the loss of neurons and progressive decline in multiple areas of functioning, including cognition, communication skills, and the ability to carry out daily activities [1]. Alzheimer’s disease (AD) is the most common of these conditions, having an accumulation of amyloid-beta protein fragments outside neurons and hyperphosphorylated tau tangles within neurons as its hallmark pathology [2]. Over 110 years ago, Alois Alzheimer first described the disease that bears his name, characterizing it by deficits in memory, impairment in verbal communication, visuospatial disorders, and changes in personality such as depression [3,4]. By 2010, 35.6 million people worldwide had dementia; 60–80% of these cases were attributed to AD. However, the most alarming aspect is that a 225% increase in the number of patients with this disease is expected worldwide by mid-century, forcing countries to allocate more resources to this population and expanding the need for more caregivers [5].

Given this scenario, emphasis has been placed on predicting who will experience AD, since an early diagnosis allows patients to enroll in clinical trials, which could help to slow the progression of the disease, better preserve cognitive functions, and provide economic and emotional benefits for both caregivers and patients [6,7,8]. For this reason, since 1988, with Barry Reisberg’s mild cognitive impairment definition (MCI), researchers have focused on distinguishing subjects with MCI who progress to AD from those who do not. MCI represents a transitional state between normal cognition and dementia, as it indicates cognitive deficits, including impairments that could be related to memory (amnestic MCI) or other cognitive abilities (non-amnestic MCI); even though not all MCI subjects progress to AD and some eventually revert to cognitive normalcy, subjects with MCI have an increased risk of developing AD [9,10].

The first criteria for diagnosing AD was created in 1984 by the National Institute of Neurological and Communicative Disorders and Stroke (NINCDS) and the Alzheimer’s Disease and Related Disorders Association (ADRDA); since then, the criteria have not changed substantially [11]. Briefly, they consist of neuropsychological tests that measure cognitive decline and symptoms of the disease, such as the Mini-Mental State Examination (MMSE) to detect cognitive decline [12], Boston Naming Test (BNT) to measure language disorders [13], Geriatric Depression Scale (GDS) to identify depression [14], and the Alzheimer’s Disease Assessment Scale-Cognitive Subscale (ADAS) to assess cognitive and non-cognitive function characteristics in people with AD [15]. In 2011, these criteria were revised due to the advances in the understanding of the disease. It was concluded that an AD diagnosis needed the same evidence as with the previous criteria, with the addition of one in five proposed biomarkers as potential support, one of them being an atrophy in the temporal lobes visualized by magnetic resonance imaging (MRI) [16].

Recently, signal- and texture-related features extracted from MRI scans and selected machine learning techniques have emerged as possible novel markers of AD [17]. In addition, studies of the progression of AD showed that highly asymmetrical contralateral hippocampi and amygdala may indicate an early and accelerated deterioration [18].

This work focuses on the study of the MCI to AD progression in the interest of achieving early detection of AD. Previously, we have proposed new biomarkers for AD from neuropsychological data, laboratory assays, and signal and texture features from T1-sequences, such as the magnetization-prepared rapid acquisition with gradient echo (MP-RAGE) [19]. Subsequently, we analyzed in a preliminary conference paper signal- and texture-related features from hippocampal T2 maps, finding 11 features significantly different between stable and non-stable MCI subjects. Volumetric information was non-significant, and all but one of the machine learning methods improved their accuracy for AD prediction by adding the signal- and texture-related features to the neuropsychological studies [20]. It is worth commenting that, to our knowledge, T2 maps have not been studied by other researchers for this purpose. Nevertheless, they have been used to detect other diseases such as hepatic fibrosis and acute or chronic heart failure [21,22].

The main objective in this study was to determine the predictive power of signal- and texture-related features extracted from T2 maps using all the 40 contralateral pairs available in ADNI images using both a univariate and a multivariate analysis between patients with MCI who progress to AD and those who remain stable.

## 2. Materials and Methods

### 2.1. Data

Data used in the preparation of this article were obtained from the ADNI database (adni.loni.usc.edu). The ADNI was launched in 2003 as a public–private partnership, led by Principal Investigator Michael W. Weiner, MD. The primary goal of ADNI has been to test whether serial MRI, Positron Emission Tomography (PET), other biological markers, and clinical and neuropsychological assessment can be combined to measure the progression of MCI and early AD. For up-to-date information, see www.adni-info.org.

Dual fast spin-echo images, one weighted to proton density (PD) and one to T2, and MP-RAGE images available up to April 2020 were retrieved from ADNI [23]. Additionally, segmentation maps for the MP-RAGE images generated through automatic whole-brain segmentations using multi-atlas propagation with enhanced registration were also downloaded [24].

### 2.2. Subject Inclusion

The experiment included only the baseline information from subjects with a baseline MCI diagnosis, between 70 and 80 years old, with available sex and years of education information and who also had the aforementioned images and segmentation map available. One subject was eliminated from the study due to poor image quality. From these, the 18 subjects who had their first AD diagnosis 2 years after their baseline visit were regarded as progressers (MCIp), while the 22 subjects who never had an AD diagnosis and participated in the study for at least 5 years were labeled as stable (MCIs). Patients who did not meet either the MCIp or the MCIs criteria were excluded from the study.

Table 1 details the demographic characteristics of the population. There was no significant difference in age and years of education between groups when tested using the Wilcoxon rank-sum test nor a significant difference in male/female proportion under a chi-squared test. MCI and AD diagnoses were determined as defined by ADNI guidelines [25].

### 2.3. MRI Processing

After the three types of images and the segmentation map were downloaded from the ADNI database for every subject, T2 maps were generated, and their 83 anatomical regions were segmented. In order to generate the T2 maps, we used the dual fast spin-echo images, namely, the PD- and T2-weighted images, each with a different echo time. As shown in (1), the T2 value for the *i*th voxel can be calculated by fitting the measured signal intensity *S* at each echo time *TE* to a mono-exponential decay function [26]:(1)$$S_{a}\left(i\right)\;=S_{0}e^{-TE_{a}/T2\left(i\right)}$$
where $S_{0}$ is the signal intensity at zero *TE*. From there, and working with the signal from the PD- and T2-weighted images ($S_{a}$ and $S_{b}$, respectively), we obtain (2)
(2)$$T2\left(i\right)\;=\frac{TE_{b}-TE_{a}}{\ln\left(S_{a}\left(i\right)\right)-\ln\left(S_{b}\left(i\right)\right)}$$
where T2(*i*) is the T2 value for the *i*th voxel, *TE_b_* and *TE_a_* represent the echo time of the T2- and PD-weighted images, respectively, and $S_{a}$(*i*) and $S_{b}$(*i*) represent the signal value of the *i*th voxel for the PD- and T2-weighted images, respectively.

To extract relevant features, it was necessary to perform a segmentation of the T2 maps. The segmentations maps downloaded from ADNI were specifically constructed for the MP-RAGE images; therefore, a registration process was required to apply these segmentation maps to the T2 maps. Spin-echo and MP-RAGE images were obtained in the same imaging session; hence, images were almost identical except for differences caused by any head movement. Using ITK [27], we performed a rigid registration between the T2 maps and their MP-RAGE counterparts using the itkVersorRigid3DTransform function with the Mattes mutual information metric, a regular step gradient descent optimizer, and a linear interpolator. Quality of the registration was confirmed visually.

### 2.4. Feature Extraction

Each anatomical region was measured for a set of 38 features: volume, 28 features related to signal distribution (e.g., energy, kurtosis, and skewness), and 9 texture-related features (e.g., mass scatter and compactness of the intensity projection map). Then, we proceeded to calculate the absolute difference and mean of each signal and texture measurement between contralateral regions. The final database consisted of the difference and mean features of 40 contralateral pairs, and 3 regions had no counterpart: brainstem (spans the midline), corpus callosum, and third ventricle. In total, there were 2960 features related to either the signal or texture of the T2 maps, plus the volume of the 83 regions.

### 2.5. Statistical Analysis

We performed a univariate and a multivariate analysis. For the former, we compared the features with the Wilcoxon rank-sum test. Then, to control the false-positive detection rate and adjust the *p*-values for multiple comparisons, a Benjamini–Hochberg procedure was used, and *q*-values were obtained [28]. A feature was determined significantly different between groups if a *q*-value lower than 0.05 was found.

We used FRESA.CAD Binary Classification Benchmarking, an R package that performs systematic comparisons between machine learning methods, to perform the multivariate analysis [29,30,31]. The methods included were: bootstrapped stage-wise model selection (BSWiMS), k-nearest neighbors (KNN) with BSWiMS features, least absolute shrinkage and selection operator (LASSO), random forest (RF), recursive partitioning and regression trees (RPART), support vector machines (SVM) with minimum-Redundancy-Maximum-Relavance (mRMR) method, and the ensemble of these methods (ENS). We performed a 100-fold cross-validation strategy, where the training sample was constructed by randomly selecting 80% of the subjects while the rest were kept for validation. For this study, we focused mainly on accuracy, sensitivity, specificity, balanced error, and the area under the receiver operating characteristic curve (ROC AUC) with a 95% confidence interval (CI).

Furthermore, in order to find the features with the highest predictive potential, we evaluated the ability of several feature-selection algorithms—integrated discrimination improvement (IDI), Kendall correlation, LASSO, mRMR, net reclassification improvement (NRI), RF, RPART, t-student test, and Wilcoxon test—in their ability to select the best set of features for several classifiers: KNN, naïve Bayes, nearest centroid with normalized root sum square distance and Spearman correlation distance, RF, and SVM. These classifiers were analyzed using the same cross-validations strategy.

### 2.6. Experiment Design

In order to find the predictive power of the features related to signal and texture, we performed two different experiments. The first one included the total scores from eight neuropsychology studies that are used for the diagnosis of AD, namely, MMSE, BNT, GDS, ADAS with 11 items (ADAS-11), and ADNI summary scores related with executive function, visuospatial functioning, language, and memory [32,33,34]. The second experiment included these 8 scores in addition to the most significant features in the univariate analysis extracted from the T2 maps.

## 3. Results

### 3.1. Univariate Analysis for Neuropsychological Studies and Volumes

The univariate analysis for the eight neuropsychological tests yielded three of them as significant: ADNI memory test (*p*-value = 7.765 × 10^−4^), ADAS-11 (*p*-value = 0.004) and MMSE (*p*-value = 0.025). Only the first two remained significant after the Benjamini–Hochberg procedure was run with the rest of the neuropsychological tests. Regarding the volumetric information, only one feature was found to be significant under the Wilcoxon rank-sum test: the right amygdala (*p*-value = 0.034).

### 3.2. Univariate Analysis for Signal and Texture Features

Of the 2960 signal and texture features, 140 were significantly different between classes, 89 mean values and 51 absolute differences. However, after adjusting for multiple comparisons using the Benjamini–Hochberg method, none of these remained significant. Table 2 shows the 25 features with the lowest *p*-values. It is worth noting that 11 of them belong to the hippocampus.

### 3.3. Multivariate Analysis for Neuropsychological Studies

The prediction results for the different machine learning techniques considering only neuropsychological tests are shown in Table 3. The features most frequently found in the predictive models were the total scores from the ADNI memory test and the ADAS-11. The machine learning technique with the best results was LASSO, with an accuracy of 0.675 and an ROC AUC of 0.727. However, it is worth noting that confidence intervals overlap, implying no real difference between methods.

### 3.4. Multivariate Analysis for Neuropsychological Studies and Imaging Features

In order to include only relevant features in the selection pool to be used for each classifier, we proceeded to take the most relevant characteristics, that is, those with the lowest *p*-value of the univariate analyses. Twelve volumes (~15%; 12/83) and 148 features related to signal and texture (~5%; 148/2960) were considered. All eight neuropsychological tests were also included.

Table 4 shows the results obtained with each of the seven machine learning techniques for the experiment with neuropsychological information, volumetric information, and signal- and texture-related information. Comparing those results with the ones found in Table 3, it can be seen that all methods had a higher average score in accuracy and ROC AUC, except for RPART. Furthermore, we can notice that the specificity, sensitivity, and balanced error were improved. Sensitivity measures the proportion of positives that are correctly identified, and specificity measures the proportion of negatives that are correctly identified. Figure 1 shows the ROC of the most relevant machine learning methods for this experiment.

As previously mentioned, we were also interested in finding out which specific features were more relevant in predicting the progression from MCI to AD. From the nine feature selection methods that were compared, the absolute difference in the Mass Scatter YY in the hippocampus, a feature related to the texture of the T2 map, was found among the six most frequent features in all of them. That is, after all feature selection methods were paired with each classifier, the frequency in which each feature was selected in the final model was computed, and this particular feature was at least the sixth most frequently selected feature every time. Similarly, the absolute difference of the value at 25% in the superior frontal gyrus, a feature related to the signal of the T2 map, was in the top-six in eight of the nine feature selection methods. Additionally, ADNI’s memory test and ADAS-11 were in the top-6 in seven and six methods, respectively. Regarding volumetric information, only in the RPART method were volumes found within the 50 most frequent features. The RPART methods used on average 9.67 features per model.

## 4. Discussion

The present study showed that the signal and texture features extracted from T2 maps could be used in conjunction with information from neuropsychological studies for the prediction of AD. To reach this conclusion, we compared the accuracy, sensitivity, specificity, balanced error, and ROC AUC for each of the different machine learning techniques between the experiment without imaging information and the one that included it. In general, and for all metrics, there was an improvement in the different techniques by adding this information. Another important aspect to highlight is that the presence of volumes in the prediction models was inconsequential.

In a review of MRI texture analyses with machine learning techniques [17], many studies performed classification and prediction of AD. Even though these studies included a greater number of subjects, the vast majority of them focused specifically on the hippocampal region and used only one machine learning technique. Additionally, they used T1-sequences, while this study focused on T2 maps of the whole brain segmented into 40 contralateral regions. Furthermore, we were able to pinpoint specific features by performing an exhaustive feature selection analysis.

The reduction in the volume of the hippocampus was one of the first biomarkers for AD classification [35]. Later, studies have reported that the texture of the hippocampus compared to its volume predicts earlier and more effectively the progression to AD [20,36,37]. In this study, the contralateral difference in a texture-related feature measured in the hippocampi was the most frequently selected feature, and several other signal- and texture-related features from the hippocampus were found to be most relevant under the univariate analysis.

However, we were able to identify other regions of the brain as potential sources for novel biomarkers of the MCI to AD progression process. For example, the contralateral difference in a signal-related feature measured in the superior frontal gyrus was the second most frequently selected feature, and that same feature yielded the lowest *p*-value when the univariate analysis was run. Similarly, signal- and texture-related features measured in the lateral ventricle, the subcallosal area, and cerebellum had some of the lowest *p*-values from the univariate analysis and were found to be frequently selected by the different feature selection methods.

This study has several limitations; for example, we only focused on 28 features related to the signal distribution and nine to the texture of the image. However, the results we obtained motivate us to follow the recommendations of The Image Biomarker Standardization Initiative [38] in search for features that can improve our models. Additionally, the inclusion criteria forced us to work with a small population, a potential cause for the lack of significant features after the *p*-value correction in the univariate analysis; we intend to run further experiments with a larger dataset derived from more relaxed inclusion criteria. Lastly, we believe this work drives further analyses and experimentation, the most important being the inclusion of information from two different MRI sequences to enhance the models.

## 5. Conclusions

T2 maps segmented into 83 anatomical brain regions from 40 subjects with MCI who either progressed to AD or remained stable were analyzed and contralateral features related to the signal and texture of the maps were extracted. We identified that the contralateral difference in a texture-related feature (the absolute differences in Mass Scatter YY) extracted from the hippocampi and the contralateral difference of a signal-related feature (the signal value at 25%) extracted from the superior frontal gyrus were the most relevant features for the task of classifying between MCIp and MCIs subjects under both a univariate and a multivariate analysis. In general, signal- and texture-related features enhanced the MCI to AD predictive power of models that used information from neuropsychological tests, such as ADAS-11 and ADNI’s memory test. Furthermore, we found that signal and texture information is more relevant for this task than mere volumetric information. These results suggest that contralateral signal- and texture-related information extracted from T2 maps should continue to be explored in the search for better MCI-to-AD predictive models.

## Figures and Tables

**Figure 1 healthcare-09-00941-f001:**
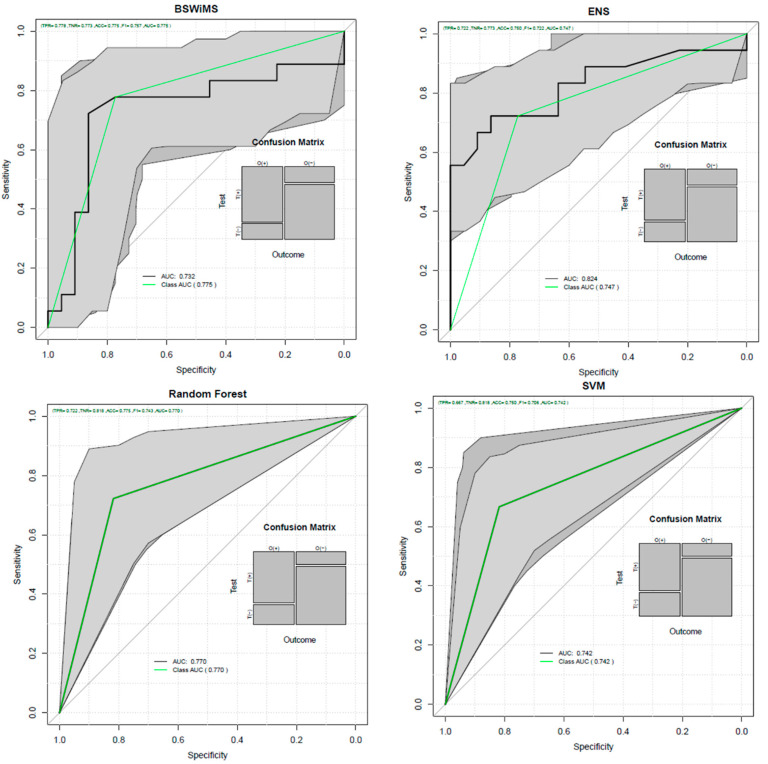
ROC AUC curves on the imaging features and neuropsychological test scores analysis.

**Table 1 healthcare-09-00941-t001:** Demography of the population.

Group of Study	Total	MCIs	MCIp	*p*-Value
Subjects (males)	40 (32)	22 (18)	18 (14)	1.000
Years of age	75.3 ± 3.0	75.3 ± 3.2	75.2 ± 2.9	0.924
Years of education	15.7 ± 3.0	15.8 ± 3.1	15.6 ± 2.9	0.879

Mean value ± standard deviation; *p*-value of the chi-squared test (male/female proportion) or Wilcoxon rank-sum test (age and education).

**Table 2 healthcare-09-00941-t002:** Significant features by their *q*-value of ${\beta_{1}^{\prime }}_{j}$.

Rank	Feature	Modality	Brain Region	*p*-Value
1	Value at 25% ^a^	Difference	Superior frontal gyrus	1.52 × 10^−4^
2	Mass Scatter YY ^b^	Difference	Hippocampus	5.51 × 10^−4^
3	σ at 90% central value ^a^	Mean	Hippocampus	0.001
4	ICV at 90% central value ^a^	Mean	Hippocampus	0.002
5	Probability of value being lower than 2σ ^a^	Difference	Lateral ventricle, temporal horn	0.002
6	Entropy ^a^	Mean	Hippocampus	0.002
7	Energy ^a^	Mean	Hippocampus	0.002
8	Value at 75% ^a^	Mean	Hippocampus	0.003
9	Skewness ^a^	Mean	Subcallosal area	0.004
10	Energy ^a^	Mean	Subcallosal area	0.004
11	Mass Scatter YY ^b^	Difference	Cerebellum	0.004
12	Value at 5% ^a^	Difference	Superior frontal gyrus	0.004
13	µ signal ^a^	Mean	Hippocampus	0.004
14	µ at 90% central value ^a^	Mean	Hippocampus	0.005
15	Entropy ^a^	Mean	Subcallosal area	0.005
16	Value at 95% ^a^	Mean	Hippocampus	0.006
17	Kurtosis ^a^	Mean	Hippocampus	0.006
18	Precision range ^a^	Mean	Hippocampus	0.006
19	Precision range ^a^	Mean	Insula	0.006
20	Value at 99.99% ^a^	Difference	Anterior orbital gyrus	0.007
21	ICV at 90% central value ^a^	Mean	Lateral occipitotemporal gyrus, gyrus fusiformis	0.008
22	Probability of value being greater than 3σ ^a^	Mean	Cingulate gyrus, posterior part	0.008
23	Energy ^a^	Mean	Cingulate gyrus, posterior part	0.008
24	Value at 25% ^a^	Difference	Putamen	0.008
25	Probability of value being greater than 3σ ^a^	Difference	Lateral ventricles, temporal horn	0.008

^a^ Features related to the signal distribution of the image; ^b^ Features related to the texture of the image.

**Table 3 healthcare-09-00941-t003:** Results for the multivariate analysis with neuropsychological tests.

	Accuracy		ROC AUC		Specificity		Sensitivity		Balanced Error	
Technique	Mean	CI	Mean	CI	Mean	CI	Mean	CI	Mean	CI
BSWIMS	0.500	0.338–0.662	0.558	0.380–0.736	0.273	0.107–0.502	0.778	0.524–0.936	0.475	0.341–0.613
ENS	0.650	0.483–0.794	0.649	0.472–0.826	0.636	0.407–0.828	0.667	0.410–0.867	0.347	0.198–0.513
KNN	0.625	0.458–0.773	0.674	0.504–0.845	0.500	0.282–0.718	0.778	0.524–0.936	0.361	0.225–0.509
LASSO	0.675	0.509–0.814	0.727	0.567–0.888	0.636	0.407–0.828	0.722	0.465–0.903	0.321	0.177–0.469
RF	0.650	0.483–0.794	0.657	0.507–0.806	0.591	0.364–0.793	0.722	0.465–0.903	0.343	0.200–0.494
RPART	0.650	0.483–0.794	0.638	0.483–0.793	0.682	0.451–0.861	0.611	0.357–0.827	0.353	0.208–0.506
SVM	0.650	0.483–0.794	0.652	0.499–0.804	0.636	0.407–0.828	0.667	0.410–0.867	0.350	0.201–0.504

**Table 4 healthcare-09-00941-t004:** Results for neuropsychological and imaging features experiment.

	Accuracy		ROC AUC		Specificity		Sensitivity		Balanced Error	
Technique	Mean	CI	Mean	CI	Mean	CI	Mean	CI	Mean	CI
BSWIMS	0.775	0.615–0.892	0.732	0.553–0.911	0.773	0.546–0.922	0.778	0.524–0.936	0.223	0.100–0.359
ENS	0.750	0.588–0.873	0.824	0.681–0.968	0.773	0.546–0.922	0.722	0.465–0.903	0.250	0.124–0.398
KNN	0.675	0.509–0.814	0.721	0.550–0.892	0.727	0.498–0.893	0.611	0.357–0.827	0.330	0.191–0.482
LASSO	0.675	0.509–0.814	0.773	0.610–0.936	0.636	0.407–0.828	0.722	0.465–0.903	0.321	0.177–0.477
RF	0.775	0.615–0.892	0.770	0.636–0.905	0.818	0.597–0.948	0.722	0.465–0.903	0.225	0.101–0.360
RPART	0.500	0.338–0.662	0.513	0.348–0.677	0.454	0.244–0.678	0.555	0.308–0.785	0.499	0.343–0.653
SVM	0.750	0.588–0.873	0.742	0.603–0.882	0.818	0.597–0.948	0.667	0.410–0.867	0.255	0.127–0.401

## Data Availability

Data used in the preparation of this article were obtained from the ADNI database (adni.loni.usc.edu, access date: 30 April 2020).

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
