# Peer review of "Signal and Texture Features from T2 Maps for the Prediction of Mild Cognitive Impairment to Alzheimer’s Disease Progression"

_healthcare, 2021, doi:10.3390/healthcare9080941_

Round 1

Reviewer 1 Report

Title: Signal and texture features from T2 maps for the prediction of Mild Cognitive Impairment to Alzheimer's disease prediction

Authors: Trejo-Castro, A.; Caballero-Luna, R.; Lopez, J.; Vega-lara, F.; Martinez-Torteya, M.

Overview: The manuscript presents the development and analysis of machine learning algorithms towards the prediction/identification of MCI cases that will progress to AD. Models were initially tested using scores of various cognitive tests. Ultimately, the texture of T2 MRI images was the primary focus of feature generation & selection; these features were added to the models. Overall, the work is interesting, and identifying persons with MCI more likely to convert to AD/dementia is important to the research and potential treatment of the disease. The work is generally well explained and well written, and the data is presented well. However, there are some errors in grammar and word choice that make certain areas more difficult to understand.

Minor Concerns:

Introduction:

  1. Minor language errors. E.g., Line 37: “For more than 110 years ago, Alois Alzheimer first described the disease thar it bears his name ...” should be changed to “more than 110 years ago, Alois Alzheimer first described the disease that now bears his name ...”
  2. Lines 45-48: very awkward wording here.
  3. Line 51: Perhaps it could be indicated that while MCI is often a transitional state as described, this is not always the case; some of those with MCI will even revert to cognitive normalcy. Furthermore, MCI is not monolithic, and subtypes such as amnestic MCI exist that exhibit different (increased in the case of amnestic) risks of conversion to AD.

Materials and Methods:

  1. What kind of cross validation was used? The sample size is small, so something like leave-one-out would work here.

Based on the description given it seems that 100 unique 80/20 train/test splits were used for each model, but this is not entirely clear.

Results:

  1. It would be interesting to see the model performance excluding the cognitive tests.

Discussion:

  1. Line 246: The “vast majority” of what?
  2. Minor language errors. E.g., line 246: “concentrated in” should be changed to “concentrated on”.
  3. Line 252: 'anticipation' is an odd word choice.
  4. Perhaps the limitations of a small sample size could be discussed.

Conclusions:

  1. Line 269: “Similarly” should be changed to something along the lines of “correspondingly”.

Reviewer 2 Report

An interesting initial investigation of novel MRI image ML for early AD diagnosis is reported by Martinez-Torteya and colleagues. I have a few specific comments for the authors outlined below.

Minor

Line 26 - “should be followed investigated as potential biomarkers…” either followed or investigated should be removed.

Line 38 - “thar” should be that

Lines 127 and 132 - Citations should be provided for both of these equations

Major

Repeated grammatical errors and spelling errors have caused me to flag editing as a topic in need of major revision in general. The sentence starting on line 45 is in need of substantial rework, for example.

Line 196 - In doing multiple univariate analyses, the authors invite the multiple comparisons problem. This limitation of the study needs to be mentioned in this section and discussed in the discussion section. Additionally, the authors need to briefly mention how future work will overcome the multiple comparisons problem (e.g. using much larger datasets at a later date that would enable a Bonferroni-corrected analysis to be undertaken).

Line 207 - The sentence starting with “The best machine learning technique” concludes that LASSO is the “best” method due to its score in accuracy and ROC. However, it can be seen that there is no statistically significant difference between the methods if one evaluates the CIs for the various methods. All of the ranges of the CIs for accuracy overlap. Therefore, the performance of the algorithms was not statistically different and no superiority claim can be made. Additionally, the authors beg the question ‘why should overall accuracy dictate superiority?’ Additional questions soon follow. If our goal is to identify the most AD patients at the earliest stage, does it not make sense to focus on sensitivity rather than accuracy? What is the harm that we could expect to befall a population by designing a sensitive method as compared to a highly specific method? These considerations need to be discussed by the authors.
